# Variational multiple shooting for Bayesian ODEs with Gaussian processes

**Pashupati Hegde**[1]   **Çağatay Yıldız**[1]   **Harri Lähdesmäki**[1]   **Samuel Kaski**[1]   **Markus Heinonen**[1]

[1] Department of Computer Science, Aalto University, Finland

## Abstract

Recent machine learning advances have proposed black-box estimation of *unknown continuous-time system dynamics* directly from data. However, earlier works are based on approximative solutions or point estimates. We propose a novel Bayesian nonparametric model that uses Gaussian processes to infer posteriors of unknown ODE systems directly from data. We derive sparse variational inference with decoupled functional sampling to represent vector field posteriors. We also introduce a probabilistic shooting augmentation to enable efficient inference from arbitrarily long trajectories. The method demonstrates the benefit of computing vector field posteriors, with predictive uncertainty scores outperforming alternative methods on multiple ODE learning tasks.

## 1 INTRODUCTION

Ordinary differential equations (ODEs) are powerful models for continuous-time non-stochastic systems, which are ubiquitous from physical and life sciences to engineering (Hirsch et al., 2012). In this work, we consider non-linear ODE systems

$$\mathbf{x}(t) = \mathbf{x}_0 + \int_0^t \mathbf{f}(\mathbf{x}(\tau))d\tau \tag{1}$$

$$\dot{\mathbf{x}}(t) := \frac{d\mathbf{x}(t)}{dt} = \mathbf{f}(\mathbf{x}(t)), \tag{2}$$

where the state vector $\mathbf{x}(t) \in \mathbb{R}^D$ evolves over time $t \in \mathbb{R}_+$ from an initial state $\mathbf{x}_0$ following its time derivative $\dot{\mathbf{x}}(t)$, and $\tau$ is an auxiliary time variable. Our goal is to learn the differential function $\mathbf{f} : \mathbb{R}^D \mapsto \mathbb{R}^D$ from state observations, when the functional form of $\mathbf{f}$ is unknown.

The conventional mechanistic approach involves manually defining the equations of dynamics and optimizing their parameters (Butcher and Goodwin, 2008), or inferring their posteriors (Girolami, 2008) from data. However, the equations are unknown or ambiguous for many systems, such as human motion (Wang et al., 2008). Some early works explored fitting unknown ODEs with splines (Henderson and Michailidis, 2014), Gaussian processes (Äijö and Lähdesmäki, 2009) or kernel methods (Heinonen and d'Alché-Buc, 2014) by resorting to less accurate gradient matching approximations (Varah, 1982). Recently, Heinonen et al. (2018) proposed estimation of free-form non-linear dynamics using Gaussian processes without gradient matching. However, the approach is restricted to learning point estimates of the dynamics, limiting the uncertainty characterization and generalization. Chen et al. (2018) proposed modeling ODEs with neural networks and adjoints, which was later extended to the Bayesian setting by Dandekar et al. (2020). However, the gradient descent training in such approaches can be ill-suited for complex or long-horizon ODEs with typically highly non-linear integration maps (Diehl and Gros, 2017).

In this work, we introduce efficient Bayesian learning of unknown, non-linear ODEs. Our contributions are:

- We introduce a way of learning posteriors of vectorfields using Gaussian processes as flexible priors over differentials $\mathbf{f}$, and thereby build on the work by Heinonen et al. (2018). We adapt decoupled functional sampling to simulate ODEs from vector field posteriors.

- For the difficult problem of gradient optimizations of ODEs, we introduce a novel probabilistic shooting method. It is motivated by the canonical shooting methods from optimal control and makes inference stable and efficient on long trajectories.

- We empirically show the effectiveness of the proposed method even while learning from a limited number of observations. We demonstrate the ability to infer arbitrarily long trajectories efficiently with the shooting extension.

*Accepted for the 38th Conference on Uncertainty in Artificial Intelligence* (UAI 2022).

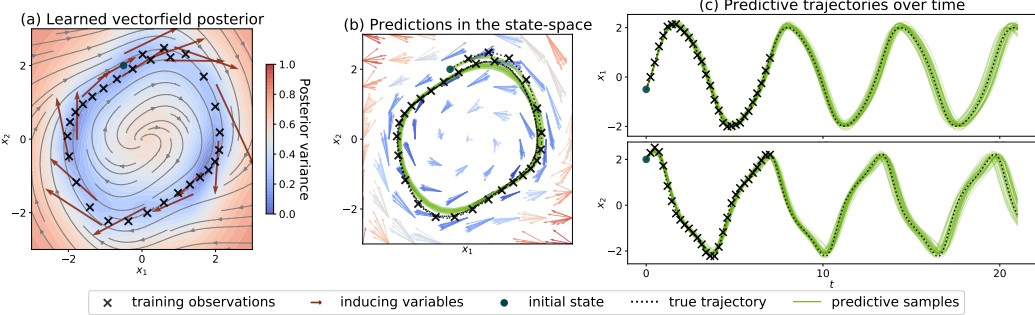

Figure 1: Illustration of GPODE: The model learns a GP posterior (a) of a vector field. Valid ODE trajectories are sampled from the posterior process as shown in (b) and (c).

## 2  RELATED WORKS

**Mechanistic ODE models.** In mechanistic modelling the equation $\mathbf{f}_\theta$ is predefined with a set of coefficients $\theta$ to be fitted (Butcher and Goodwin, 2008). Several works have proposed embedding mechanistic models within Bayesian or Gaussian process models (Calderhead et al., 2008; Dondelinger et al., 2013; Wenk et al., 2020). Recently both Julia and Stan have introduced support for Bayesian analysis of parametric ODEs (Rackauckas and Nie, 2017; Stan, 2021). Since this line of work assumes a known dynamics model, we do not consider these methods in the experiments.

**Free-form ODE models.** Multiple works have proposed fitting unknown, non-linear and free-form ODE differentials with gradient matching using splines (Ramsay et al., 2007), Gaussian processes (Äijö and Lähdesmäki, 2009) or kernel methods (Heinonen and d'Alché-Buc, 2014). Recently, Heinonen et al. (2018) proposed accurate *maximum a posteriori*(MAP) optimisation of vector fields with sensitivity equation gradients (Kokotovic and Heller, 1967). Neural ODEs (Chen et al., 2018) introduced adjoint gradients (Pontryagin et al., 1962) along with flexible black-box neural network vector fields. Several extensions to learning latent ODEs have been proposed (Yildiz et al., 2019; Rubanova et al., 2019).

**Discrete-time state-space models.** There is a large literature on Markovian state-space models that operate over discrete time increments (Wang et al., 2005; Turner et al., 2010; Frigola et al., 2014). Typically nonlinear state transition functions are modeled with Gaussian processes and applied to latent state estimation or system identification problems with dynamical systems (Eleftheriadis et al., 2017; Doerr et al., 2018; Ialongo et al., 2019). In this paper, we focus strictly on continuous-time models and leave the study of discrete vs. continuous formulations for future work.

**Stochastic differential equations.** As an alternative formulation of inferring unknown dynamics from observational data, one can assume stochastic transitions and learn models of stochastic differential equations (SDEs). Existing works have utilized Gaussian processes (Archambeau et al., 2007; Duncker et al., 2019; Jørgensen et al., 2020) and neural networks (Tzen and Raginsky, 2019; Li et al., 2020) to model non-linear SDEs. However, since they assume a different model (i.e. deterministic transitions vs stochastic transitions), we will restrict the experimental comparisons to other ODE-based approaches.

## 3  METHODS

We consider the problem of learning ODEs (2) with GPs and propose a Bayesian model to infer posteriors over the differential $\mathbf{f}(\cdot)$.

### 3.1  BAYESIAN MODELING OF ODES USING GPS

We assume a sequence of $N$ observations $\mathbf{Y} = (\mathbf{y}_1, \mathbf{y}_2, \ldots \mathbf{y}_N)^T \in \mathbb{R}^{N \times D}$ along a trajectory, with $\mathbf{y}_i \in \mathbb{R}^D$ representing the noisy observation of the unknown state $\mathbf{x}(t_i) \in \mathbb{R}^D$ at time $t_i$. Similar to Heinonen et al. (2018), we assume a zero mean vector-valued Gaussian process prior over $\mathbf{f}$,

$$\mathbf{f}(\mathbf{x}) \sim \mathcal{GP}(\mathbf{0}, K(\mathbf{x}, \mathbf{x}')), \qquad (3)$$

which defines a distribution of differentials $\mathbf{f}(\mathbf{x})$ with covariance $\mathrm{cov}[\mathbf{f}(\mathbf{x}), \mathbf{f}(\mathbf{x}')] = K(\mathbf{x}, \mathbf{x}')$, where $K(\mathbf{x}, \mathbf{x}') \in \mathbb{R}^{D \times D}$ is a stationary matrix-valued kernel. We follow the commonly used sparse inference framework for GPs using inducing variables (Titsias, 2009), and augment the full model with inducing values $\mathbf{U} = (\mathbf{u}_1, \ldots, \mathbf{u}_M)^T \in \mathbb{R}^{M \times D}$ and inducing locations $\mathbf{Z} = (\mathbf{z}_1, \ldots, \mathbf{z}_M)^T \in \mathbb{R}^{M \times D}$ such that $\mathbf{u}_m = \mathbf{f}(\mathbf{z}_m)$. The inducing variables are trainable 'landmark' state-differential pairs, from which the rest of the differential field is interpolated (See Figure 1, where arrow locations are the $\mathbf{z}_m$ and arrow end-points are the $\mathbf{u}_m$). The inducing augmentation leads to the following prior and

conditionals (Hensman et al., 2013):

$$p(\mathbf{U}) = \mathcal{N}(\mathbf{U}|\mathbf{0}, \mathbf{K_{ZZ}}), \tag{4}$$

$$p(\mathbf{f}|\mathbf{U}; \mathbf{Z}) = \mathcal{N}(\mathbf{f}|\mathbf{A}\text{vec}(\mathbf{U}), \mathbf{K_{XX}} - \mathbf{A}\mathbf{K_{ZZ}}\mathbf{A}^T), \tag{5}$$

where $\mathbf{X} = (\mathbf{x}_1, \mathbf{x}_2, \dots \mathbf{x}_{N'})^T \in \mathbb{R}^{N' \times D}$ collects all the intermediate state evaluations $\mathbf{x}(t_i)$ encountered along a numerical approximation of the true continuous ODE integral (1), $\mathbf{f} = (\mathbf{f}(\mathbf{x}_1)^T, \dots, \mathbf{f}(\mathbf{x}_{N'})^T)^T \in \mathbb{R}^{N'D \times 1}$, $\mathbf{K_{XX}}$ is a block-partitioned matrix of size $N'D \times N'D$ with $D \times D$ blocks, so that block $(\mathbf{K_{XX}})_{i,j} = K(\mathbf{x}_i, \mathbf{x}_j)$, and $\mathbf{A} = \mathbf{K_{XZ}}\mathbf{K_{ZZ}}^{-1}$. For notational simplicity, we assume that the measurement time points are among the time points of the intermediate state evaluations of a numerical ODE solver.

The joint probability distribution follows

$$p(\mathbf{Y}, \mathbf{f}, \mathbf{U}, \mathbf{x}_0) = p(\mathbf{Y}|\mathbf{f}, \mathbf{x}_0)p(\mathbf{f}, \mathbf{U})p(\mathbf{x}_0) \tag{6}$$

$$= \prod_{i=1}^{N} p(\mathbf{y}_i|\mathbf{f}, \mathbf{x}_0)p(\mathbf{f}|\mathbf{U})p(\mathbf{U})p(\mathbf{x}_0), \tag{7}$$

where the conditional distribution $p(\mathbf{y}_i|\mathbf{f}, \mathbf{x}_0) = p(\mathbf{y}_i|\mathbf{x}_i)$ computes the likelihood over ODE state solutions $\mathbf{x}_i = \mathbf{x}_0 + \int_0^{t_i} \mathbf{f}(\mathbf{x}(\tau))d\tau$.

## 3.2   VARIATIONAL INFERENCE FOR GP-ODES

In contrast to earlier approach that estimates MAP solutions (Heinonen et al., 2018), our goal is to infer the posterior distribution $p(\mathbf{f}, \mathbf{x}_0|\mathbf{Y})$ of the vector field $\mathbf{f}$ and initial value $\mathbf{x}_0$ from observations $\mathbf{Y}$. The posterior is intractable due to the non-linear integration map $\mathbf{x}_0 \xmapsto{\mathbf{f}} \mathbf{x}(t)$.

We use the stochastic variational inference (SVI) formulation for sparse GPs (Hensman et al., 2013) in this work. We introduce a factorized Gaussian posterior approximation for the inducing variables across state dimensions $q(\mathbf{U}) = \prod_{d=1}^{D} \mathcal{N}(\mathbf{u}_d|\mathbf{m}_d, \mathbf{Q}_d), \mathbf{u}_d \in \mathbb{R}^M$ where $\mathbf{m}_d \in \mathbb{R}^M, \mathbf{Q}_d \in \mathbb{R}^{M \times M}$ are the mean and covariance parameters of the variational Gaussian posterior approximation for the inducing variables. We treat the inducing locations $\mathbf{Z}$ as optimized hyperparameters. The posterior distribution for the variational approximation can be written as

$$q(\mathbf{f}) = \int p(\mathbf{f}|\mathbf{U})q(\mathbf{U})d\mathbf{U} \tag{8}$$

$$= \int \mathcal{N}\left(\mathbf{f}|\mathbf{A}\text{vec}(\mathbf{U}), \mathbf{K_{XX}} - \mathbf{A}\mathbf{K_{ZZ}}\mathbf{A}^T\right) q(\mathbf{U})d\mathbf{U}. \tag{9}$$

The posterior inference goal then translates to estimating the posterior $p(\mathbf{f}, \mathbf{U}, \mathbf{x}_0|\mathbf{Y})$ of the inducing points $\mathbf{U}$ and initial state $\mathbf{x}_0$. Under variational inference this learning objective

$$\underset{q}{\arg\min} \; \text{KL}\left[ q(\mathbf{f}, \mathbf{U}, \mathbf{x}_0) \,||\, p(\mathbf{f}, \mathbf{U}, \mathbf{x}_0|\mathbf{Y}) \right] \tag{10}$$

translates into maximizing the evidence lowerbound (ELBO),

$$\log p(\mathbf{Y}) \geq \sum_{i=1}^{N} \overbrace{\mathbb{E}_{q(\mathbf{f}, \mathbf{x}_0)} \log p(\mathbf{y}_i|\mathbf{f}, \mathbf{x}_0)}^{\text{variational likelihood}} - \overbrace{\text{KL}[q(\mathbf{U})||p(\mathbf{U})]}^{\text{inducing KL}}$$

$$- \underbrace{\text{KL}[q(\mathbf{x}_0)||p(\mathbf{x}_0)]}_{\text{initial state KL}}, \tag{11}$$

where we also assume variational approximation $q(\mathbf{x}_0) = \mathcal{N}(\mathbf{a}_0, \Sigma_0)$ for the initial state $\mathbf{x}_0$. See supplementary section 1.1 for detailed derivations of the above equations.

## 3.3   SAMPLING ODES FROM GAUSSIAN PROCESSES

The Picard-Lindelöf theorem (Lindelöf, 1894) ensures valid ODE systems define unique solutions to the initial value problem (IVP) (1). In order to sample valid state trajectories for the IVP, we need to efficiently sample GP functions $\mathbf{f}(\cdot) \sim q(\mathbf{f})$ (9). This way, we can evaluate the sample function $\mathbf{f}(\mathbf{x}(t))$ at arbitrary states $\mathbf{x}(t)$ encountered during ODE forward integration, while accounting for both the inducing and interpolation distributions of Equation (9). Unfortunately, function-space sampling of such GPs has prohibitive cubic complexity (Rasmussen and Williams, 2006; Ustyuzhaninov et al., 2020), while the more efficient weight-space sampling with Fouriers cannot accurately express the posterior (9) (Wilson et al., 2020).

We use the decoupled sampling that decomposes the posterior into two parts (Wilson et al., 2020),

$$\overbrace{\mathbf{f}(\mathbf{x})|\mathbf{U}}^{\text{posterior}} = \overbrace{\mathbf{f}(\mathbf{x})}^{\text{prior}} + \overbrace{K(\mathbf{x}, \mathbf{Z})K(\mathbf{Z}, \mathbf{Z})^{-1}(\mathbf{U} - \mathbf{f_Z})}^{\text{update}}. \tag{12}$$

$$\approx \sum_{i=1}^{F} \mathbf{w}_i \phi_i(\mathbf{x}) + \sum_{j=1}^{M} \boldsymbol{\nu}_j K(\mathbf{x}, \mathbf{z}_j), \tag{13}$$

where we use $F$ Fourier bases $\phi_i(\cdot)$ with $\mathbf{w}_i \sim \mathcal{N}(\mathbf{0}, I)$ (Rahimi and Recht, 2007) to represent the stationary prior, and function basis $K(\cdot, \mathbf{z}_j)$ for the posterior update with $\boldsymbol{\nu} = K(\mathbf{Z}, \mathbf{Z})^{-1}(\mathbf{U} - \boldsymbol{\Phi}\mathbf{W}), \boldsymbol{\Phi} = \phi(\mathbf{Z}) \in \mathbb{R}^{M \times F}, \mathbf{W} \in \mathbb{R}^{F \times D}$. By combining these two steps, we can accurately evaluate functions from the posterior (9) in linear time at arbitrary locations. We refer the reader to the supplementary section 1.2 for more details. We note that concurrent works by Mikheeva et al. (2021) and Ensinger et al. (2021) also utilize the decoupled-sampling to infer ODE posteriors with GPs.

## 3.4   AUGMENTING THE ODE MODEL WITH SHOOTING SYSTEM

A key bottleneck in ODE modeling is the poor gradient descent performance over long integration times $\mathbf{x}_{0:T}$, which

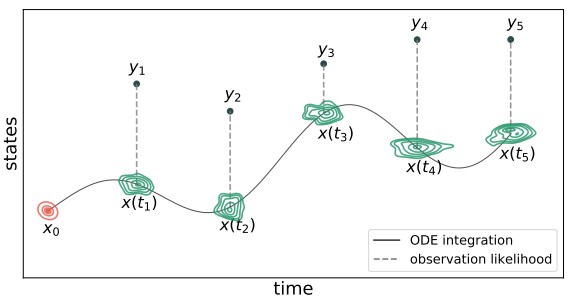

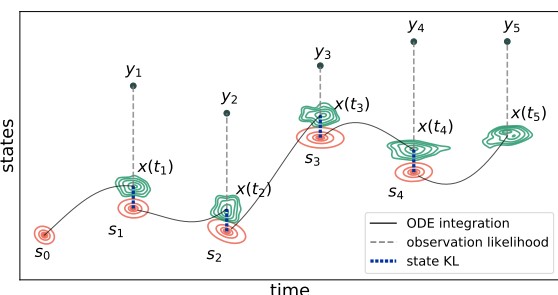

| (a) The full model formulation | (b) Shooting augmentation |

Figure 2: Illustrations of GPODE formulations: the full model formulation (a) follows the long trajectory integration, whereas the shooting version (b) splits the long trajectory into multiple short subintervals.

can exhibit vanishing or exploding gradients (Haber and Ruthotto, 2017; Choromanski et al., 2020). Earlier approaches tackled this issue mainly with more accurate numerical solvers (Zhuang et al., 2020, 2021). The nonlinearity of the integration map $\mathbf{x}_0 \overset{\mathbf{f}}{\mapsto} \mathbf{x}_t$ motivates us to instead segment the full integration $\mathbf{x}_{0:T}$ into short segments, which are easier to optimize and can be trivially parallelized. This is called the *multiple shooting* method in optimal control literature (Osborne, 1969; Bock and Plitt, 1984), in the context of parameter estimation of ODEs (vanDomselaar and Hemker, 1975; Bock, 1983). Recently, Massaroli et al. (2021); Turan and Jäschke (2021) also introduced a multiple-shooting framework within the context of deterministic neural ODEs. We introduce probabilistic shooting for the Gaussian process posterior inference of ODEs.

We begin by introducing shooting state variables $\mathbf{S} = (\mathbf{s}_0, \mathbf{s}_1, \ldots, \mathbf{s}_{N-1})$, $\mathbf{s}_i \in \mathbb{R}^D$, and segment the continuous state function $\mathbf{x}(t; \mathbf{x}_0)$ (1) into $N$ segments $\{(\mathbf{s}_{i-1}, \mathbf{x}(t_i; \mathbf{s}_{i-1}))\}_{i=1}^N$ that branch from the shooting variables $\mathbf{s}_{i-1}$ (See Figure 2);

$$\mathbf{x}(t_i; \mathbf{s}_{i-1}) = \mathbf{s}_{i-1} + \int_{t_{i-1}}^{t_i} \mathbf{f}(\mathbf{x}(\tau))d\tau. \quad (14)$$

In addition, every shooting variable is approximately matched with the ODE state evolution from the previous shooting state,

$$\mathbf{s}_i = \mathbf{x}(t_i; \mathbf{s}_{i-1}) + \boldsymbol{\xi}, \quad (15)$$

where $\boldsymbol{\xi} \in \mathbb{R}^D$ represents the tolerance parameter controlling the shooting approximation. The augmented system is equivalent to the original ODE system in case the constraints $\mathbf{s}_i = \mathbf{x}(t_i; \mathbf{s}_{i-1})$ are satisfied exactly at the limit $\boldsymbol{\xi} \to \mathbf{0}$. We place a Gaussian prior over the tolerance parameter $\boldsymbol{\xi} \sim \mathcal{N}(\mathbf{0}, \sigma_\xi^2 \mathbf{I})$, which translates into the following prior over shooting variables

$$p(\mathbf{s}_i | \mathbf{s}_{i-1}) = \mathcal{N}(\mathbf{s}_i | \mathbf{x}(t_i; \mathbf{s}_{i-1}), \sigma_\xi^2 \mathbf{I}). \quad (16)$$

Further, the joint probability of the augmented model after

placing a GP prior over the vectorfield $\mathbf{f}$ can be written as

$$p(\mathbf{Y}, \mathbf{S}, \mathbf{f}) = \prod_{i=1}^N p(\mathbf{y}_i | \mathbf{s}_{i-1}, \mathbf{f}) \prod_{i=1}^{N-1} p(\mathbf{s}_i | \mathbf{s}_{i-1}, \mathbf{f}) p(\mathbf{s}_0) p(\mathbf{f}). \quad (17)$$

### 3.5 VARIATIONAL INFERENCE FOR THE AUGMENTED MODEL

To infer the augmented posterior $p(\mathbf{f}, \mathbf{U}, \mathbf{S} | \mathbf{Y})$ we introduce variational approximation for the shooting variables $q(\mathbf{S}) = q(\mathbf{s}_0) \cdots q(\mathbf{s}_{N-1})$, where each distribution $q(\mathbf{s}_i) = \mathcal{N}(\mathbf{s}_i | \mathbf{a}_i, \Sigma_i)$ is a Gaussian. This results in the joint variational approximation

$$q(\mathbf{S}, \mathbf{f}, \mathbf{U}) = \prod_{i=0}^{N-1} q(\mathbf{s}_i) p(\mathbf{f} | \mathbf{U}) q(\mathbf{U}), \quad (18)$$

and the following evidence lower bound for the shooting model,

$$\mathcal{L}_{\text{shooting}} = \sum_{i=1}^N \mathbb{E}_{q(\mathbf{s}_{i-1}, \mathbf{f})} \Big[ \log p(\mathbf{y}_i | \mathbf{s}_{i-1}, \mathbf{f}) \Big]$$

$$+ \sum_{i=1}^{N-1} \mathbb{E}_{q(\mathbf{s}_i, \mathbf{s}_{i-1}, \mathbf{f})} \Big[ \log p(\mathbf{s}_i | \mathbf{s}_{i-1}, \mathbf{f}) \Big] - \mathbb{E}_{q(\mathbf{s}_i)} \Big[ \log q(\mathbf{s}_i) \Big]$$

$$- \text{KL}[q(\mathbf{s}_0) \,\|\, p(\mathbf{s}_0)] - \text{KL}[q(\mathbf{U}) \,\|\, p(\mathbf{U})]. \quad (19)$$

The ELBO consists of an expected log-likelihood term, which matches the state evolution (14) from every shooting variable to the corresponding observation. In addition, the posterior approximation for every shooting variable is also matched with the ODE evolution of the approximated posterior of the previous shooting state, leading to corresponding cross-entropy and entropy terms.

The ELBO for the augmented shooting model requires solving only the short segments (14) with simpler integration maps, thus great at mitigating problems with vanishing/exploring gradients. Since the involved numerical ODE

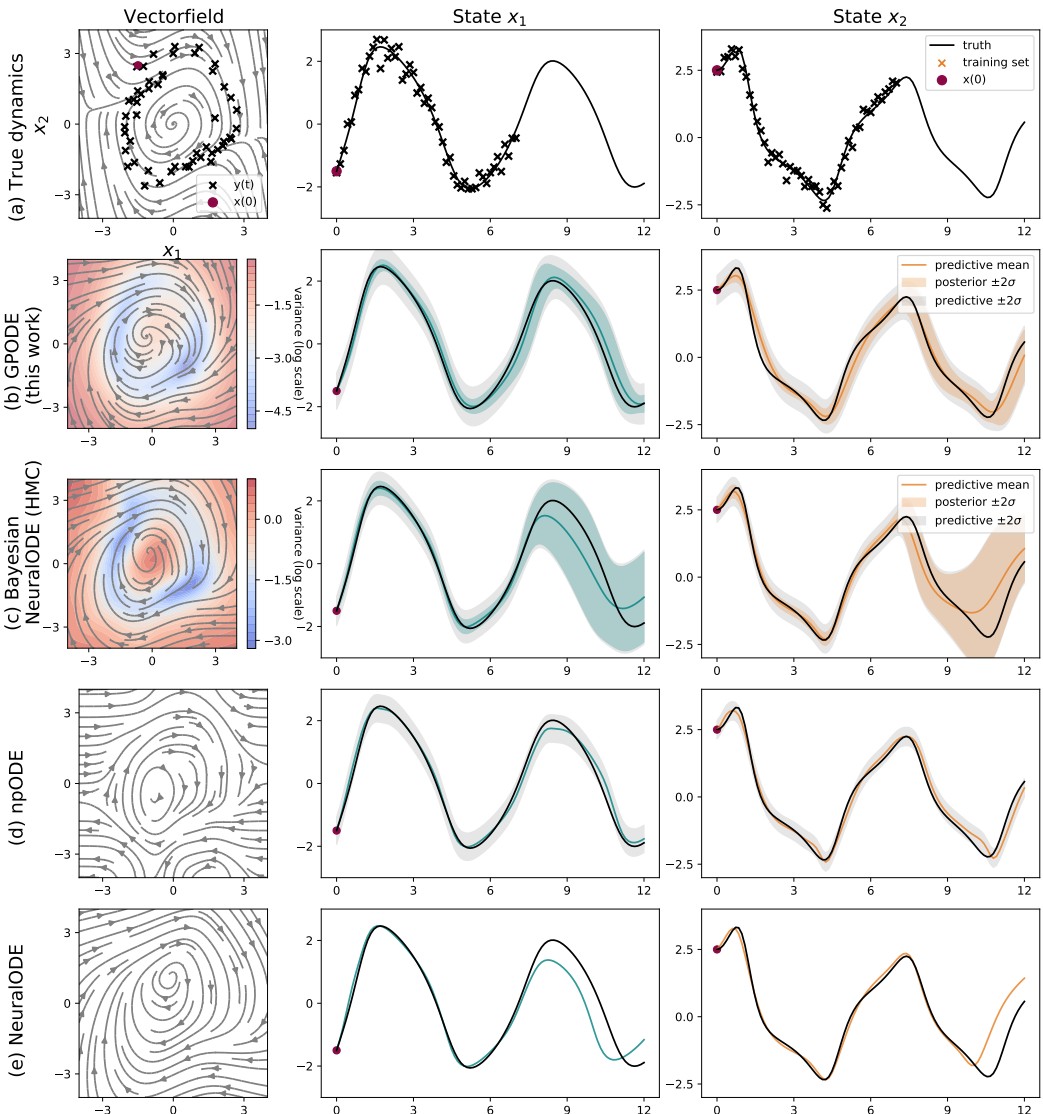

Figure 3: Learning the 2D Van der Pol dynamics **(a)** with alternative methods **(b-d)**. Column 1 shows the vector fields while columns 2 and 3 show the state trajectories $x_1(t)$ and $x_2(t)$. GPODE learns the posterior accurately.

integration can be done in parallel, the shooting model is also computationally faster than the full model in practice. See supplementary section 1.3 for a plate diagram and detailed derivation of the approach.

## 4 EXPERIMENTS

We validate the proposed method on Van der Pol (VDP) and FitzHugh–Nagumo (FHN) systems and on the task of learning human motion dynamics (MoCap). The predictive performance of the proposed GPODE is compared against npODE (Heinonen et al., 2018), NeuralODE (Chen et al., 2018) and Bayesian version of NeuralODE (Dandekar et al., 2020). We use 16 inducing points in VDP and FHN experiments and 100 inducing points for the MoCap experiments.

Except for the NeuralODE model, we assume Gaussian observation likelihood, and infer the unknown noise scale parameter from the training data. All the experiments use squared exponential kernel with automatic relevance determination (ARD) along with 256 Fourier basis functions for decoupled GP sampling. Along with the variational parameters, kernel lengthscales, signal variance, noise scale, and inducing locations are jointly optimized against the model ELBO while training. In addition, for the shooting model, we fix the constraint tolerance parameter to a small value $\sigma_\xi^2 = 1e^{-6}$ consistently across all the experiments. In all the shooting experiments, we considered the number of shooting segments to be the same as the number of observation segments in the dataset. A codebase for implementing the proposed methods is provided `https://github.com/hegdepashupati/gaussian-process-odes`.

Table 1: VDP system learning performance on extrapolation task with observations on regular (task 1) and irregular time intervals (task 2). We report mean ± standard error over 5 runs from different random initialization, the best values bolded. (↑): higher is better, (↓) lower is better

| | Task 1: Regular time-grid | | Task 2: Irregular time-grid | |
|---|---|---|---|---|
| | MNLL (↓) | MSE (↓) | MNLL (↓) | MSE (↓) |
| Bayesian NeuralODE (HMC) | $0.82 \pm 0.01$ | $1.45 \pm 0.04$ | $0.88 \pm 0.01$ | $1.68 \pm 0.04$ |
| NeuralODE | - | $0.29 \pm 0.11$ | - | $0.55 \pm 0.07$ |
| npODE | $1.47 \pm 0.59$ | $0.16 \pm 0.05$ | $8.89 \pm 3.06$ | $2.08 \pm 0.78$ |
| GPODE | $\mathbf{0.60 \pm 0.03}$ | $\mathbf{0.13 \pm 0.01}$ | $\mathbf{0.41 \pm 0.18}$ | $\mathbf{0.21 \pm 0.07}$ |

We use the `dopri5` solver with tolerance parameters `rtol=` $1e^{-5}$ and `atol=` $1e^{-5}$, and use the adjoint method for computing loss gradients with `torchdiffeq`[1] package (Chen et al., 2018). All the experiments are repeated 5 times with random initialization, and means and standard errors are reported over multiple runs. The predictive performance of different models are measured with mean squared error (MSE) and mean negative log likelihood (MNLL) metrics.

### 4.1 LEARNING VAN DER POL DYNAMICS

We first illustrate the effectiveness of the proposed method by inferring the vector field posterior on a two-dimensional VDP (see Figure 3),

$$\dot{x}_1 = x_2, \qquad (20)$$
$$\dot{x}_2 = -x_1 + 0.5x_2(1 - x_1^2).$$

We simulate a trajectory of 50 states following the true system dynamics from the initial state $(x_1(0), x_2(0)) = (-1.5, 2.5)$, and add Gaussian noise with $\sigma^2 = 0.05$ to generate the training data. We explore two scenarios with training time interval $t \in [0, 7]$ and forecasting interval $t \in [7, 14]$: (1) over a regularly sampled time grid, (2) over an irregular grid using uniform random sampling of time points. Task (2) demonstrates one of the key advantages of continuous-time models with the ability to handle irregular data.

Figure 3(b) shows that both GPODE and Bayesian NeuralODE learn a vector field posterior whose posterior mean closely matches the ground truth, with low variance (blue regions) near the observed data. The posterior variance increases away from the observed data (orange regions), indicating a good uncertainty characterization, while the npODE with MAP estimation seems to overfit. NeuralODE learns an appropriate vector field, but requires careful tuning of regularization and hyperparameters for a good fit with a limited number of observations. A quantitative evaluation of the model fits in Table 1 indicates the better performance

[1]https://github.com/rtqichen/torchdiffeq

Table 2: Imputation results on the FHN system.

| | MNLL (↓) | MSE (↓) |
|---|---|---|
| Bayesian NeuralODE (HMC) | $0.77 \pm 0.12$ | $0.24 \pm 0.03$ |
| NeuralODE | - | $0.18 \pm 0.00$ |
| npODE | $6.49 \pm 1.49$ | $\mathbf{0.08 \pm 0.01}$ |
| GPODE | $\mathbf{0.09 \pm 0.05}$ | $\mathbf{0.07 \pm 0.02}$ |

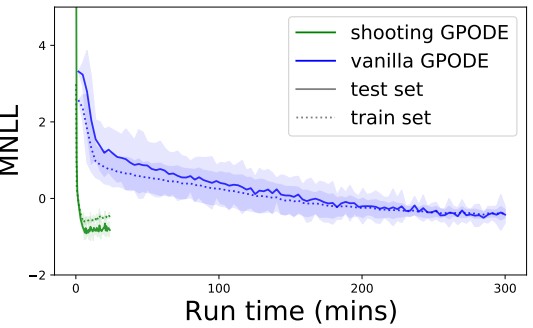

Figure 4: Optimization efficiency with GPODE models.

of GPODE as compared to the other methods under comparison.

### 4.2 LEARNING WITH MISSING OBSERVATIONS

We illustrate the usefulness of learning Bayesian ODE posteriors under missing data with the FHN oscillator

$$\dot{x}_1 = 3(x_1 - x_1^3/3 + x_2), \qquad (21)$$
$$\dot{x}_2 = (0.2 - 3x_1 - 0.2x_2)/3.$$

We generate a training sequence by simulating 25 regularly-sampled time points from $t \in [0, 5.0]$ with added Gaussian noise with $\sigma^2 = 0.025$. We remove all observations at the quadrant $x_1 > 0, x_2 < 0$ and evaluate model accuracy in this region. The interpolation performance for different models is shown in Table 2. The point estimates of npODE and NeuralODE have biases, while the Bayesian variants of GPODE and NeuralODE provide good uncertainty estimates corresponding to their better predictive performance.

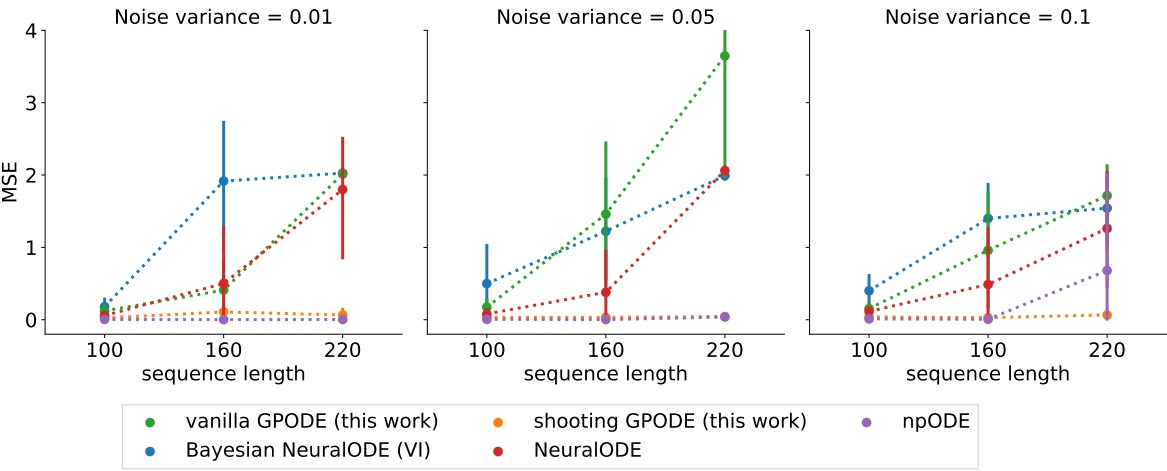

Figure 5: Varying sequence length and observation noise: shooting formulation makes GPODE feasible for long sequences, outperforming the non-shooting version and competing methods. We report the results for different levels of observation noise and training sequence length on the VDP system.

## 4.3 LEARNING LONG TRAJECTORIES WITH THE SHOOTING FORMULATION

We demonstrate the necessity of the shooting formulation for working with long training trajectories. We use the VDP system with four observations per unit of time for $T = (25, 40, 55)$ corresponding to $N = (100, 160, 220)$ observed states. We also vary the observation variance as $\sigma^2 = (0.01, 0.05, 0.1)$ and test the model for forecasting additional 50 time points.

Figure 5 demonstrates that vanilla-GPODE and NeuralODE, and Bayesian NeuralODE fail to fit the data with long sequences on all noise levels. In contrast, inference for the shooting model is successful in all settings. The npODE is remarkably robust to long trajectories. We believe the robustness of npODE mainly stems from the excellent parameter initialization strategy (see supplementary section 2.2) coupled with the fully deterministic optimization setup (no reparametrization gradients).

Figure 4 shows a runtime trace comparison between vanilla GPODE and the shooting variant in wall-clock time for a fixed budget of 15000 optimization steps on the VDP system with $N = 100$, $T = 25$ and $\sigma^2 = 0.01$. The shooting model converges approximately 10 times faster. The speedup stems from the parallelization of the shooting ODE solver, since the shooting method splits the full IVP problem into numerous short and less non-linear IVPs. In addition, the shooting method relaxes the inference problem with its auxiliary augmentation. This experiment was conducted on a system with AMD Ryzen 5 3600 processor and Nvidia GeForce GTX 1660S GPUs.

## 4.4 LEARNING HUMAN MOTION DYNAMICS

We learn the dynamics of human motion from noisy experimental data from CMU MoCap database for three subjects, `09`, `35` and `39`. The dataset consists of 50 sensor readings from different parts of the body while walking or running. We follow the preprocessing of Wang et al. (2008) and center the data. The dataset was further split into train, test, and validation sequences. We observed that the NeuralODE, the Bayesian NeuralODE version with VI, and npODE models suffer from over-fitting, and we remedy this by applying early stopping by monitoring the validation loss during optimization.

We project the original 50-dimensional data into a 5-dimensional latent space using PCA and learn the dynamics in the latent space. To compute the data likelihood, we project the latent dynamics back to the original data space by inverting the PCA. We divide the experiment into sub-tasks MoCap-short and MoCap-long, based on the length of the sequence considered for model training (see the supplementary section for more details on the dataset and experimental setup). The model predictive performance is measured on unseen test sequences in both tasks.

Table 3 indicates that GPODE outperforms the competing npODE and NeuralODE model variants. Figure 6 visualizes the predicted dynamics for a test sequence. The GPODE variants have reasonable posterior uncertainties, while NeuralODE variants and npODE tend to be overconfident and make more mistakes (see Figure 6 (b), sensors `05`, `41` and `47`) . We note that some variations in the data space cannot be accurately estimated due to the low-dimensional PCA projection.

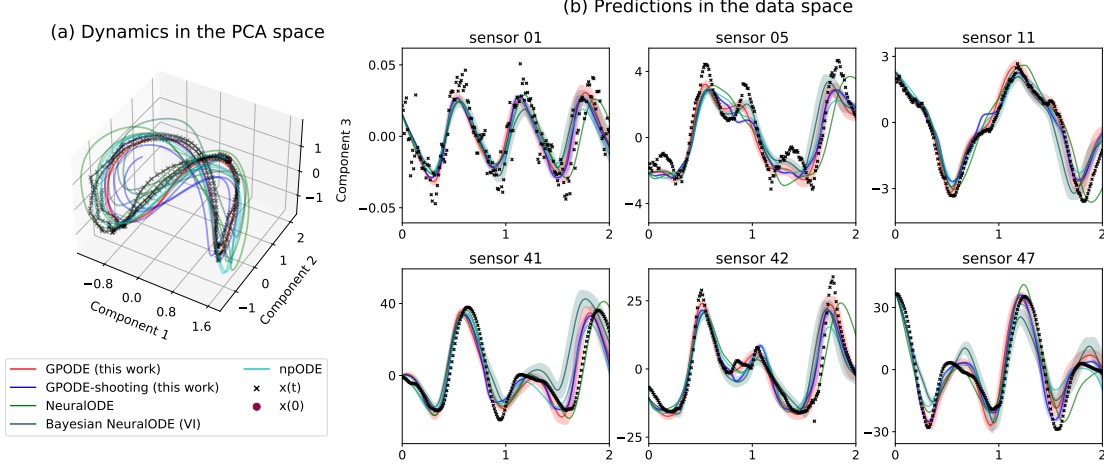

Figure 6: Learning the walking dynamics of subject `39`: The true dynamics and predicted dynamics (mean) for the first three components in PCA space are shown in (a). Corresponding trajectories in the observation space for 6 different sensors are shown in (b) (We do not plot the observation noise variance)

Table 3: Test MNLL and MSE metrics for dynamics prediction task on CMU MoCap dataset.

| Metric | Method | Subject 09 | | Subject 35 | | Subject 39 | |
|---|---|---|---|---|---|---|---|
| | | short | long | short | long | short | long |
| MNLL(↓) | Bayesian NeuralODE (VI) | $2.03 \pm 0.10$ | $1.50 \pm 0.05$ | $1.42 \pm 0.05$ | $1.37 \pm 0.06$ | $1.61 \pm 0.07$ | $1.45 \pm 0.03$ |
| | npODE | $2.09 \pm 0.01$ | $1.78 \pm 0.08$ | $1.67 \pm 0.02$ | $1.66 \pm 0.04$ | $2.06 \pm 0.05$ | $1.78 \pm 0.04$ |
| | GPODE-vanilla | $1.30 \pm 0.02$ | $1.26 \pm 0.02$ | $1.27 \pm 0.04$ | $1.39 \pm 0.04$ | $1.29 \pm 0.01$ | $\mathbf{1.13 \pm 0.01}$ |
| | GPODE-shooting | $\mathbf{1.19 \pm 0.02}$ | $\mathbf{1.14 \pm 0.02}$ | $\mathbf{1.25 \pm 0.06}$ | $\mathbf{1.08 \pm 0.02}$ | $1.25 \pm 0.01$ | $1.36 \pm 0.02$ |
| MSE(↓) | Bayesian NeuralODE (VI) | $25.50 \pm 1.70$ | $21.32 \pm 2.58$ | $23.09 \pm 3.95$ | $20.86 \pm 2.95$ | $53.34 \pm 5.31$ | $39.66 \pm 6.82$ |
| | NeuralODE | $27.53 \pm 2.87$ | $33.83 \pm 2.46$ | $36.50 \pm 3.86$ | $23.54 \pm 0.56$ | $115.38 \pm 10.96$ | $53.51 \pm 2.98$ |
| | npODE | $17.91 \pm 1.62$ | $19.76 \pm 4.29$ | $26.24 \pm 2.88$ | $22.83 \pm 3.91$ | $92.80 \pm 15.74$ | $55.94 \pm 4.63$ |
| | GPODE-vanilla | $15.78 \pm 0.67$ | $12.62 \pm 1.14$ | $16.14 \pm 0.99$ | $15.53 \pm 0.76$ | $\mathbf{20.71 \pm 1.25}$ | $23.64 \pm 1.94$ |
| | GPODE-shooting | $\mathbf{9.11 \pm 0.37}$ | $\mathbf{8.38 \pm 1.23}$ | $\mathbf{10.11 \pm 0.79}$ | $\mathbf{11.66 \pm 0.73}$ | $26.72 \pm 0.63$ | $\mathbf{21.17 \pm 2.88}$ |

## 5 CONCLUSION AND DISCUSSION

We proposed a novel model for Bayesian inference of ODEs using Gaussian processes. With this approach, one can model unknown ODE systems directly from the observational data and learn posteriors of the continuous-time vector fields. In contrast, earlier works produce point estimate solutions. We believe this to be a significant addition to the data-descriptive ODE modeling methods, especially for applications where uncertainty quantification is critical. Many conventional machine learning algorithms have been interpreted and modeled as continuous-time dynamical systems, with applications to generative modeling (Grathwohl et al., 2019) and probabilistic alignment (Ustyuzhaninov et al., 2020), among others. However, scaling GPs to high-dimensional datasets (such as images) can be a bottleneck. The applicability of the proposed model as a plug-in extension for these applications can be studied as part of future work.

We also highlighted a problem of learning black-box ODE models on long trajectories and proposed a probabilistic shooting framework enabling efficient inference on such

tasks. This framework can be applied to other existing approaches, such as NeuralODEs. However, the proposed shooting augmentation introduces model approximation and involves approximating inference over auxiliary shooting variables. Hence the benefits of the shooting augmentation can be task specific, especially on short sequences. Comprehensive empirical studies across different types of tasks can be considered in future work.

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
