# OpenReview forum: "Variational multiple shooting for Bayesian ODEs with Gaussian processes"
_auai.org/UAI/2022/Conference — UAI 2022 Oral_

### Official Review · Reviewer_ADoW · 2022-04-13

**Q2(1) Originality/Novelty:** 3
**Q2(2) Significance/Impact:** 3
**Q2(3) Correctness/Technical Quality:** 3
**Q2(6) Clarity Of Writing:** 4
**Q6 Overall Score:** 8
**Q8 Confidence In Your Score:** 4

**Q1 Summary And Contributions:**

The authors describe a method of learning a vector field from samples along differential equation trajectories.  They take a variational inference approach using GPs as the trial distributions, and learn based on the behavior of integrations along segments of the trajectory together with matching conditions (with noise) to tie them together, similar to multiple shooting methods used in two-point boundary value problems.

**Q2 Assessment Of The Paper:**

More detailed information regarding each of these aspects is given below:

**Q2(4) Quality Of Experiments (Optional):**

3: Good: The experimental evaluation is adequate, and the results convincingly support the main claims.

**Q2(5) Reproducibility:**

3: Good: Key resources (e.g., proofs, code, data) are available and key details (e.g., proofs, experimental setup) are sufficiently well-described for competent researchers to confidently reproduce the main results.

**Q3 Main Strengths:**

Multiple shooting is a good idea for boundary value problems and optimal control, and it is a good idea here.  The variational inference framework also seems like the right way to go for this problem.

**Q4 Main Weakness:**

It would have been nice to see more discussion of the impacts of the variational inference approximation on the behavior, but this is a quibble.  I see no significant weaknesses.  I like this paper a good deal.

**Q5 Detailed Comments To The Authors:**

The gradient matching approach seems a little unfairly maligned here, and I think a version might be worth thinking about even if you do not intend to pursue it in the long term.  Using the data to fit a smoothing spline in the case of the VDP example, for instance, would give state-plus-gradient estimates (with noise) that could be treated as direct *linear* observations of the GP in either a full inference or a VI procedure.

Particularly in low-dimensional spaces, the cubic scaling of direct solvers for the full GP is a bit of a red herring -- fast approximation methods (SKI or inducing point methods) seem like they should be easily within reach, and I would imagine that one of these methods together with a smoothing spline for gradient estimation would be significantly faster than the proposed VI-based GPODE methods (even the one with shooting).  If the accuracy of the gradients is really a serious issue, this might also solve as a useful initialization for the VI-based GPODE solver.

**Q7 Justification For Your Score:**

I like this paper a great deal.  I think it's a good idea in other settings and a good idea here, with demonstrated impact on training times.

**Q9 Complying With Reviewing Instructions:**

1: Yes.

---

### Official Review · Reviewer_9qbu · 2022-04-14

**Q2(1) Originality/Novelty:** 3
**Q2(2) Significance/Impact:** 3
**Q2(3) Correctness/Technical Quality:** 3
**Q2(6) Clarity Of Writing:** 3
**Q6 Overall Score:** 7
**Q8 Confidence In Your Score:** 4

**Q1 Summary And Contributions:**

This paper aims to learn posteriors for continuous-time ODE systems using Gaussian processes priors. For inference, it proposes to first use the stochastic variational inference formulation to approximate the posterior with sparse Gaussian processes and then utilize decoupled sampling to sample from the posterior approximation. In order to alleviate the poor performance on long trajectory learning, a shooting system is further proposed to combine with the proposed Bayesian learning.

**Q2 Assessment Of The Paper:**

More detailed information regarding each of these aspects is given below:

**Q2(4) Quality Of Experiments (Optional):**

4: Excellent: The experimental evaluation is comprehensive and the results are compelling.

**Q2(5) Reproducibility:**

2: Fair: Key resources (e.g., proofs, code, data) are unavailable but key details (e.g., proof sketches, experimental setup) are sufficiently well-described for an expert to confidently reproduce the main results.

**Q3 Main Strengths:**

The task of learning continuous-time ODE is important and it is specifically interesting to do it in a Bayesian setting where the uncertainty of the learned solution is quantified. The proposed Bayesian learning scheme with variational inference and decoupled sampling is an effective solution. The shooting system is novel to me and it is able to generate more reliable solutions in an efficient way than the proposed one without a shooting system as well as other baselines, as indicated by the empirical evaluations.


**Q4 Main Weakness:**

This work covers a broad range of techniques including Bayesian learning using Gaussian processes, stochastic variational inference, decoupled sampling, and shooting systems. The presentation of this work could be further improved if background introductions to these techniques are included before going into the details of the proposed algorithm.

**Q5 Detailed Comments To The Authors:**

- In Fig. 4, it seems that the shooting GPOSE has not converged yet. Why not show the full curve until it converges?

- In Table 2, why the MNLL result for NeuralODE is missing?

- In the experiments in Fig 5, I would expect the average MSE would increase as the noise variance increases. However, it doesn't seem to be the case as presented in Fig 5 as the case with a noise variance of 0.05 has the highest average MSE. Any explanations on why?

- Minor issues:
	- Below Eq (5), it should be f(x_{N'}) instead of f(x'_N)
	- f_Z in Eq (12) is not defined.
	- ARD at the beginning of Sec. 4 is not defined.


**Q7 Justification For Your Score:**

The contribution of this work seems solid to me, albeit with some minor issues.

**Q9 Complying With Reviewing Instructions:**

1: Yes.

---

### Official Review · Reviewer_yGNu · 2022-04-19

**Q2(1) Originality/Novelty:** 2
**Q2(2) Significance/Impact:** 3
**Q2(3) Correctness/Technical Quality:** 3
**Q2(6) Clarity Of Writing:** 4
**Q6 Overall Score:** 7
**Q8 Confidence In Your Score:** 4

**Q1 Summary And Contributions:**

The paper proposes a variational inference-based GP formulation for ODE modeling. Sampling from the proposed model is made more efficient by utilizing the idea of multiple shooting for the proposed method. The method is then evaluated on synthetic and real-world data.

**Q2 Assessment Of The Paper:**

More detailed information regarding each of these aspects is given below:

**Q2(4) Quality Of Experiments (Optional):**

3: Good: The experimental evaluation is adequate, and the results convincingly support the main claims.

**Q2(5) Reproducibility:**

3: Good: Key resources (e.g., proofs, code, data) are available and key details (e.g., proofs, experimental setup) are sufficiently well-described for competent researchers to confidently reproduce the main results.

**Q3 Main Strengths:**

- The paper is well written and easy to follow
- Techniques used to obtain the final method are sensible choices

**Q4 Main Weakness:**

- Overall the experiments are fine, but some additional insights and discussion of them would be welcomed.

**Q5 Detailed Comments To The Authors:**

The paper is well written and easy to follow. The idea of the proposed method is sensible and an interesting approach in the area of ODE learning. From my understanding, the paper combines existing ideas in an interesting and logical manner.

The use of SVI to obtain posterior estimates and integrating multiple shooting to address numerical issues and gain performance all seem like good choices. While it does not seem there is novelty in the techniques used per se, their combination and derivation/integration required to combine them is a valid and novel contribution.
An aspect of Figure 2 that had me confused is that the shooting starting locations are significantly different from the observed/predicted locations. Is this difference as stark in real scenarios or just in the figure for visual purposes?

The experimental section showcases the benefits of the proposed method. One shortcoming of the experiments is that there is little discussion or insights about the results themselves. For example, npODE performs quite well in Figure 5, but no hypothesis as to why is provided. Similar on the MoCap dataset, the proposed system has its performance decreased by half on Subject 39, yet its MNLL score remains on the same level as for the other subjects. What is the reason for this?

The addition of the runtime information is a welcome addition. As far as I could tell, the information regarding the number of splits used was not mentioned, which would be a good addition.

**Q7 Justification For Your Score:**

The proposed method is a sound combination and extension of ideas used in related domains. The description and writing are clear and easy to follow. The experiments adequately evaluate the method and contrast it with related methods. The paper could be improved by providing a more detailed discussion of the results.

**Q9 Complying With Reviewing Instructions:**

1: Yes.

---

### Official Review · Reviewer_yWgH · 2022-04-19

**Q2(1) Originality/Novelty:** 3
**Q2(2) Significance/Impact:** 3
**Q2(3) Correctness/Technical Quality:** 3
**Q2(6) Clarity Of Writing:** 3
**Q6 Overall Score:** 7
**Q8 Confidence In Your Score:** 4

**Q1 Summary And Contributions:**

This paper presents a method for solving free-form non-linear ordinary differential equations in dynamical systems via a variational Gaussian processes framework. The method applies multiple shooting, which consists of approximately sampling from the GP posterior to solve the variational inference problem for non-overlapping segments of state trajectories.

**Q2 Assessment Of The Paper:**

More detailed information regarding each of these aspects is given below:

**Q2(4) Quality Of Experiments (Optional):**

3: Good: The experimental evaluation is adequate, and the results convincingly support the main claims.

**Q2(5) Reproducibility:**

3: Good: Key resources (e.g., proofs, code, data) are available and key details (e.g., proofs, experimental setup) are sufficiently well-described for competent researchers to confidently reproduce the main results.

**Q3 Main Strengths:**

The problem of solving non-linear ODEs in settings without much prior knowledge other than observational data is quite relevant to a variety of applications, especially in the life sciences. This paper provides a solution to this problem applying GPs, which have been well studied in the literature and provide robust theoretical guarantees, combining them with recent advancements that allow for computationally efficient posterior sampling. Experiments show predictive performance improvements over previous methods, such as the well known Neural ODEs, and exact inference via Hamiltonian Monte Carlo on Bayesian Neural ODEs. Therefore, I believe the paper provides an important contribution to the machine learning community.

**Q4 Main Weakness:**

1. Some methodological details have been left unclear in the main text. Sec. 3 did not explain how the multiple-shooting approach in Sec. 3.3 plays a role in the ELBO calculation in Sec. 3.5. These details are only found in the appendix.
2. Lack of discussion of drawbacks in design choices. The variational approximation makes each intermediate state $\mathbf s_i$ independent of each other in a type of mean-field approach. Consequences of this approximation are not discussed.
3. Experiments are missing an important detail on sampling. I missed the information on the number of samples $S$ for multiple shooting that are applied in the experiments.  Multiple shooting is one of the main components of the proposed methodology, and I believe the number of samples might play a very significant role in the methods approximation qualities.

**Q5 Detailed Comments To The Authors:**

In addition to the points Q4, I leave the following questions:
1. Any comments on the drawbacks of the mean-field variational approximation in this framework, especially in the context of predicting long sequences which may not have a strong periodic component?
2. Have there been any ablation experiments relating the number of shooting samples to the resulting model's predictive performance?

**Q7 Justification For Your Score:**

I tend to vote for acceptance, though I still have a few minor concerns that I believe the paper needs to address in its revision.

**Q9 Complying With Reviewing Instructions:**

1: Yes.

---

### Decision · Program_Chairs · 2022-05-15

**Decision:**

Accept (Oral)

**Comment:**

Meta Review: The paper addresses the important problem of inferring continuous-time ODE systems with Gaussian processes and develop an efficient multiple shooting scheme to tackle long trajectories. The reviewers were all very positive about the paper and commended the technical advances it proposes as well as the solid results. In the final version, please address the small issues to improve clarity, particularly experimental details.